# Toxigenic Cyanobacteria and Microcystins in a Large Northern Oligotrophic Lake Onego, Russia

**DOI:** 10.3390/toxins16110457

**Published:** 2024-10-25

**Authors:** Elena Tekanova, Sergey Sidelev, Nataliia Kalinkina, Ekaterina Chernova, Sophia Barinova, Andrey Sharov, Valeria Smirnova

**Affiliations:** 1Karelian Research Center, Russian Academy of Sciences, 185030 Petrozavodsk, Russia; etekanova@mail.ru (E.T.); sidelev@mail.ru (S.S.); cerioda@mail.ru (N.K.); smirnovalera24@yandex.ru (V.S.); 2Faculty of Biology and Ecology, P.G. Demidov Yaroslavl State University, 150057 Yaroslavl, Russia; 3Scientific Research Centre for Ecological Safety, St. Petersburg Federal Research Center, Russian Academy of Sciences, 197110 St. Petersburg, Russia; s3561389@yandex.ru; 4Institute of Evolution, University of Haifa, Haifa 3498838, Israel; 5Papanin Institute for Biology of Inland Waters, Russian Academy of Sciences, Borok, 152742 Yaroslavl, Russia; sharov@ibiw.ru

**Keywords:** Lake Onego, water bloom, cyanobacteria, microcystins, toxigenic species

## Abstract

Toxigenic cyanobacteria and microcystins in the oligotrophic pelagic zone and mesotrophic bay of Lake Onego—the second largest lake in Europe—were found for the first time. Microscopic analysis revealed that *Dolichospermum lemmermannii, D. circinale* and *D. spiroides* dominated in bloom spots in the oligotrophic zone of the lake and *D. flos-aquae* and *Microcystis aeruginosa* OKin the eutrophic bay. The abundance of cyanobacteria in bloom spots is potentially hazardous for humans and animals. PCR-analysis showed that *mcyA* gene involved in microcystin biosynthesis was found in cyanobacteria of the genera *Dolichospermum* and *Microcystis*. Five structural variants of intracellular microcystins were detected in a trace amount using high-performance liquid chromatography–mass-spectrometry of high resolution. The most hazardous hepatotoxin, MC-LR, was found only in the eutrophic bay. In the present study, the reasons for the low cyanotoxin content in the phytoplankton dominated by *Dolichospermum* are discussed. The findings of our study make a significant contribution to the accumulation of facts which state that toxigenic cyanobacterial blooms can occur in large oligotrophic lakes.

## 1. Introduction

Cyanobacterial blooms are a well-studied natural phenomenon [1,2,3,4]. The extensive growth of cyanobacteria takes place during the seasonal succession of phytoplankton in calm weather, when the water is well-heated, and nutrients are abundant. The danger of the phenomenon is associated not only with the degradation of aquatic ecosystems (trophic links are simplified, organic matter is accumulated, and the shortage of dissolved oxygen is felt), but also with the invasion of toxic cyanobacteria. Toxin-producing species, belonging to the genera *Microcystis*, *Cylindrospermopsis* (*Raphidiopsis*), *Dolichospermum*, *Aphanizomenon*, *Planktothrix*, *Nodularia*, etc., are known [2,4,5,6,7]. The most common and hazardous cyanobacterial toxins are hepatotoxic microcystins (MCs) [5].

In oligotrophic, especially northern, water bodies, where the abundance of cyanobacteria is usually limited by nutrients and low water temperature, the formation of bloom spots has been a rare phenomenon. However, due to climate warming, water blooms have spread to northern lakes, where this phenomenon had not previously been observed [3,8,9,10]. Water blooming in large oligotrophic lakes with significant freshwater reserves and biological resources has been a global challenge. Microcystins can have adverse effects on humans through the consumption of poorly treated drinking water, as well as on aquatic biota, including fish [11,12,13,14]. Hepatotoxins can bioaccumulate in fish directly when released into the water, indirectly through food chains, or be absorbed through the gills [11,12,13,14]. Toxic cyanobacterial blooms have been shown to cause significant economic costs [15].

Information on cyanotoxin occurrence in the water bodies of Northwest Russia is missing. Cyanotoxin presence has been studied in the Gulf of Finland of the Baltic Sea [16], in lakes on the Karelian Isthmus [17], and in a number of lakes, such as Krivoye, Krugloye [18], Sestroretsky Razliv, Nizhneye Suzdalskoye [19], and Chudsko-Pskovskoye [20]. The phenomenon of cyanobacteria blooming in Lake Onego was documented for the first time in July 2022 [21]. This phenomenon requires study of its possible causes and an assessment of its harmfulness.

The aim of the present study was to obtain the first data on the occurrence and distribution of potentially toxic cyanobacteria and microcystins in the large oligotrophic Lake Onego.

## 2. Results

Bloom spots in the pelagic zone of the oligotrophic Lake Onego were observed in July 2022. The total abundance and biomass of cyanobacteria in the pelagic zone were abnormally high for these oligotrophic parts (Table 1). High concentrations of chlorophyll *a* in water samples (31–47 µg L^−1^) corresponded with the eutrophic state (Table 1). Surface bloom scum consisted almost entirely of *Dolichospermum* species (Table 2). *Dolichospermum lemmermannii* (P.G.Richt) P.Wacklin, L.Hoffm. and J.Komárek was one of the dominant species in all three stations (S_1, S_3, L_5). *Dolichospermum spiroides* (Kleb.) (P.Wacklin, L.Hoffm. and J.Komárek) was a co-dominant species at station S_1, and *Dolichospermum circinale* (Rabenh. ex Bornet and Flahault) P.Wacklin, L.Hoffm. and J.Komárek) at stations S_3 and L_5. Other cyanobacterial taxa did not contribute markedly to the formation of bloom spots in the pelagic part of the lake, except for sub-domination in abundance of *Aphanothece nidularis* P.G.Richt at station L_5.

Cyanobacterial abundance and biomass were lower in Kondopozhskaya Bay compared to the pelagic zone by 2–10 and 10–50 times, respectively. In Kondopozhskaya Bay, the concentration of chlorophyll a was also 3–4 times lower, which was associated with the recorded low abundance and biomass (Table 1). *Dolichospermum flos-aquae* (Brébisson ex Bornet and Flahault) P.Wacklin, L.Hoffm. and J.Komárek, *D. spiroides* and *Microcystis aeruginosa* Kütz.gure were dominant species in the mesotrophic Kondopozhskaya Bay (station K50). *D. flos-aquae* made up over half of the cyanobacterial community in abundance and biomass. *D. lemmermannii*, prevailing in the pelagic zone, was a sub-dominant species in the Bay (Table 2).

In our study, extracellular MCs were not detected. Intracellular MCs were recorded at two stations, K50 and S_3 (Table 3). At station K50, MC concentration was the highest (153 ng L^−1^), and 5 structural variants [D-Asp^3^]MC-LR, MC-LR, [D-Asp^3^]MC-RR, MC-RR, MC-YR] were identified (Figure 1 and Figure 2). The share of [D-Asp^3^]MC-RR of the total content was maximal (up to 82%). The share of the most toxic variant of MC-LR was only 5% of the total. At station S_3, MC concentration was lower (12 ng L^−1^); two structural variants (MC-YR and [D-Asp^3^]MC-LR) were identified (Table 3; Figure 1). For reliable identification, the fragment spectra were obtained from the detected MC congeners (Figure 3). The characteristic fragments for each congener were selected according to the data on MC fragmentation pattern obtained using the mass spectrometer LTQ Orbitrap XL (Thermo Fisher Scientific, Waltham, MA, USA), reported in our previous work [22]. They were as follows: *m*/*z* 599 [Arg-Adda-Glu+H]^+^ and *m*/*z* 499 [Ala-Arg-D-Asp-Arg+H]^+^ for [D-Asp^3^]MC-RR; *m*/*z* 599 [Arg-Adda-Glu+H]^+^ and *m*/*z* 539 [Mdha-Ala-Leu-D-Asp-Arg+H]+ for [D-Asp^3^]MC-LR; *m*/*z* 596 [Mdha-Ala-Arg-MeAsp-Arg+H]^+^ and *m*/*z* 453 [Arg-Adda+H−NH_3_]^+^ for MC-RR; *m*/*z* 599 [Arg-Adda-Glu+H]^+^ and *m*/*z* 553 [Mdha-Ala-Leu-MeAsp-Arg+H]^+^ for MC-LR; *m*/*z* 599 [Arg-Adda-Glu+H]^+^ and *m*/*z* [Mdha-Ala-Tyr-MeAsp-Arg+H]^+^ for MC-YR.

Since *Dolichospermum* spp. are known to produce anatoxin-a [23], screening for the presence of anatoxin-a was performed using high-resolution mass spectrometry. Signals corresponding to anatoxin-a (*m*/*z* 166.12318) were not detected in any samples studied. An example of an isolated ion mass chromatogram is shown in Figure 3. However, an intense signal for the isobaric compound phenylalanine (*m*/*z* 166.0860) was present in the mass chromatogram for each phytoplankton sample examined (Figure 3A). This signal could be misidentified as anatoxin-a. The difference between these *m*/*z* values could only be distinguished using high-resolution mass spectrometry. In the case of low-resolution, fragment spectra should be examined for reliable identification [24].

PCR-analysis performed with genus-specific primers *mcyA*_MF/MR revealed amplification products of about 225 bp length in eDNA from stations K50, S_1 and S_3 (Table 3, Appendix A). The PCR product, 225 bp in length, was also amplified using the genus-specific primers *mcyA*_AF/AR in eDNA samples collected at stations K50 and S_3 but were not found in eDNA sample taken at station S_1 (Table 3, Appendix A).

## 3. Discussion

Toxigenic cyanobacteria and microcystins in the oligotrophic pelagic zone and mesotrophic bay of Lake Onego were found for the first time, which raises serious concerns. Lake Onega is a source of drinking water for the population of three large cities. The lake’s ichthyofauna includes valuable fish species (salmon and whitefish), so industrial and amateur fishing is developed. The lake is a major aquaculture site, where trout are grown. The detection of MC-producing cyanobacteria may pose potential risks of harmful effects on human health and aquatic fauna.

Microscopic analysis showed that the surface scum consisted almost entirely of *Dolichospermum* species with a minor contribution of other cyanobacteria, such as *M. aeruginosa*. Four species of *Dolichospermum* were previously identified in the phytoplankton of Lake Onego, where *D. lemmermannii* and *D. flos-aquae* have been most common [25,26]. The growth level of the two other *Dolichospermum* spp., which inhabited Lake Onego, *D. spiroides* and *D. circinale*, was much lower; and they occurred as dominant only in the shallow littoral and skerry zones of the lake [26,27,28]. The seasonal maximum of *Dolichospermum* in Lake Onego usually occurs in late July–August in a period of maximal water warming. Until now, the abundance of *D. flos-aquae* and *D. lemmermannii* in the pelagic zone did not exceed 10^3^ cells L^−1^ [25,26]. In our study, the recorded cyanobacteria abundance in the samples from the pelagic zone of Lake Onego exceeded 20–100 × 10^6^ cells L^−1^ (Table 2). In accordance with the WHO guidelines, the recorded abundance of cyanobacteria is associated with an expected high concentration of MC, which may pose a risk to human health [29].

Analysis of the cyanobacterial composition of the bloom spots indicated *Dolichospermum* and *Microcystis* as likely producers of MCs. Molecular analysis was performed to search for the *mcyA* gene of MC biosynthesis. Fragments of *mcyA* gene specific to *Microcystis* were found in eDNA from three collected samples (S_1, S_3, K50). This confirmed the potential ability of *Microcystis* to produce MCs. Microscopic analysis revealed the presence of *M. aeruginosa* colonies in lake phytoplankton at stations K50 and S_3. MCs were detected in the same samples by HPLC–MSHR. At station S_1, neither *M*. *aeruginosa* nor MCs were found. The discrepancy of the results for station S_1 obtained by different methods could be explained by the high sensitivity of the PCR method [30,31]. Baker et al. (2002) [32] showed that 10 cells mL^−1^ is a sufficient population density of toxigenic cyanobacteria for the successful amplification of the *mcyA* gene from eDNA. Fragments of the *mcyA* gene specific to *Dolichospermum* were detected in eDNA extracted from the plankton samples K50 and S_3. Thus, PCR-analysis indicated that *Dolichospermum* species were potential producers of MCs in Lake Onego. In our study, four *Dolichospermum* species were identified, and three of them (*D. flos-aquae*, *D. circinale* and *D. lemmermannii*) are known as MC producers [33,34,35]. In sample S_1, despite *D. lemmermannii* being identified, *Dolichospermum*-specific *mcyA* in eDNA and cell-bound MCs were absent (Table 3). Apparently, the non-MC-producing population of *D. lemmermannii* was present at station S_1. At stations K50 and S_3, *D*. *lemmermannii* was present together with other possible MC producers, such as *D. flos-aquae* and *D. circinale* (Table 2). We assumed that these two co-dominated *Dolichospermum* species could produce MCs. In the future, for more accurate identification of MC producers at the species level, it is necessary to isolate strains.

In spite of the significant abundance of cyanobacteria and proved presence of microcystin-producing species, MCs were detected in surface bloom scum only at station S_3, and their concentration was at trace level. In mesotrophic Kondopogskaya Bay (station K50), chlorophyll *a*, cell density and biomass of cyanobacteria were much lower than in the pelagic zone (Table 2) and recorded intracellular MC concentration did not exceed 0.15 µg L^−1^. We hypothesize that bloom spots formed in the coastal zone and in the bays can subsequently be carried by the current to the pelagic zone. Moreover, it is known that Lake Onego has a system of stable and vigorous circulation currents [36]. A similar mechanism of surface scum formation in the pelagic zone has been noted in other large oligotrophic lakes. For instance, a bloom spot of *Dolichospermum* in Lake Superior moved over a distance of up to 8 km from the shore [37].

In general, MC concentrations in bloom spots were one order of magnitude lower than guideline values (1 and 24 µg L^−1^, accordingly) established by WHO for drinking and recreational waters [38]. We did not expect significant toxic risks to human health from consumption of lake fish due to the low concentrations of MC in the bloom spots. Recent studies have not found bioaccumulation of MC in fish meat that would be hazardous to humans in large eutrophic lakes with prolonged periods of cyanobacterial blooms [39,40]. Nevertheless, obtaining data on MC content in fish from Lake Onego is important, but is beyond the scope of this study.

The low intracellular MC content in surface scum dominated by *Dolichospermum* could be due to a low proportion of MC-producing genotypes related to total cell numbers. Earlier, a similar observation was made in some lakes in Germany and Denmark, where it has been demonstrated that the MC content of *Dolichospermum*–dominated samples was lower than that of *Microcystis-* and *Planktothrix*-dominated samples [1]. Probably, trace MC concentrations in bloom spots could have been provoked by a high abundance of akinetes found in the filaments and colonies of *Dolichospermum* species. The number of akinetes in individual species was not calculated, but they generally make up 50–60% of the total number of *Dolichospermum* cells in the pelagic zone and about 30% in Kondopogskaya Bay. It has been known that akinetes are used by cyanobacteria for withstanding unfavorable conditions, such as low water temperature or lack of nutrients. In particular, limited phosphate supply is recognized as a trigger of akinete differentiation in oligotrophic water bodies [41,42]. We assumed, that akinete formation of *Dolichospermum* in bloom spots in July 2022 seemed to have been supported by phosphorus deficit in the water of the pelagic zone, where TP concentration is lower than 5–10 µg L^−1^ [43]. *Dolichospermum* akinetes were found to contain genes responsible for the biosynthesis of neurotoxic anatoxin-a [44]. However, whether MCs and other cyanotoxins are retained in akinetes after its formation has not yet been made clear. If MCs are present in small amounts in akinetes, or if they are absent, then trace intracellular concentrations of this cyanotoxin in bloom spots in Lake Onego might be explained.

According to the published data, *Dolichospermum* contributes substantially to water bloom and MC occurrence in large deep oligotrophic lakes. The growth of *Dolichospermum* in deep oligotrophic lakes is favored by its buoyancy, the formation of heterocysts for nitrogen fixation, and survival supported by akinetes. The presence of gas vesicles and a buoyancy control mechanism enables *Dolichospermum* to migrate within a large photic zone in oligotrophic lakes to search for optimum photic and trophic conditions and to spread over great distances [7,37,45]. *D. lemmermannii* is a major contributor to harmful blooms in the near-shore zone of Lake Superior. Bloom occurred at maximum seasonal water temperatures of 19.5–23.5 °C in August, and recorded MC concentrations were less than 0.1 µg L^−1^ [37]. In Lake Baikal, water blooms recorded in the near-shore zone were affected by human activities. During the bloom, *Dolichospermum* and *Microcystis* dominated; and extracellular MC concentrations varied from 0.1 to 0.17 µg L^−1^. Intracellular MC content cannot be compared with our data, because the MC concentration in biomass was expressed in µg g^−1^ dry weight [46,47]. The mono-domination of *D. sigmoideum* (Nygaard) Wacklin, L. Hoffmann and Komárek was noted in Lake Ladoga in the Valaam Archipelago in July 2021 with an abundance of 145.6 × 10^6^ cells L^−1^ and a biomass of 9.53 mg L^−1^, but no MCs were detected. Later in August, cyanobacteria were represented by *Dolichospermum* and *Aphanizomenon flos-aquae*. The abundance of 614.0 × 10^6^ cells L^−1^ and a biomass of 58.37 mg L^−1^ were recorded, and intracellular MC concentration was 7.71 µg L^−1^, including 0.18 µg L^−1^ MC-LR (Unpublished results). In Lake Michigan, subjected to reoligotrophization, water blooms were documented in the eutrophicated Green Bay in July–August. Average MC concentrations produced by *Dolichospermum* and *Microcystis* varied from 0.9 to 5 µg L^−1^ [48]. Thus, we can summarize that MC concentrations are not high in large oligotrophic lakes even when cyanobacterial bloom occurs. This evidence is in good agreement with the first data on MC concentrations in Lake Onego obtained during our study.

## 4. Conclusions

This paper presents the first results on the occurrence of MCs and their producers in registered bloom spots in the large oligotrophic Lake Onego in July 2022. It was shown that cyanobacterial communities consisted almost entirely of *Dolichospermum* spp., while *M. aeruginosa* was present in small amounts in bloom spots in the oligotrophic pelagic zone (South Onego and Maloe Onego). Molecular–genetic analysis has supported the potential ability of *Dolichospermum* and *Microcystis* to produce MCs. The recorded abundance of cyanobacteria in bloom spots in the pelagic zone could be potentially hazardous to man and animals. However, MC concentration was at trace level in the cyanobacterial biomass from bloom spots. We believe that this could be due to either a very low abundance of MC-producing genotypes among cyanobacterial populations or an abundance of akinetes in a potentially toxigenic *Dolichospermum*. It should be emphasized that MC content in cyanobacterial akinetes has not yet been thoroughly studied.

The findings of our study confirmed the information in the literature on the possibility of cyanobacterial blooms in large oligotrophic lakes, which is important for understanding the mechanisms of eutrophication of large lakes and developing a theory of eutrophication.

## 5. Materials and Methods

### 5.1. General Characteristics of the Study Area

Lake Onego (61°38′34″ N 35°31′08″ W), the second largest body of water in Europe, is located in northwestern Russia on the border with the subarctic zone. The lake is of glacial origin. Its northern part is located on the Fennoscandian crystalline shield, and the southern part is on the East European Plain. The area of the lake is 9720 km^2^, the water volume is 295 km^3^, the average depth is 30 m, and the maximum is 120 m. The period of conditional water exchange is 15.6 years. The reservoir has 152 tributaries, including 52 rivers longer than 10 km. The Svir River flows out of the lake. The catchment area is 70% covered by coniferous forests, and 17% by open bogs [49]. In spring and autumn, the water is completely mixed. The average surface temperature of Lake Onego during the ice-free period (190 days) does not exceed 8 °C. The duration of biological summer with a water temperature above 10 °C is 100 days [36]. Lake Onego is a low-mineralized oligotrophic reservoir (Table 4). The polluted part of the lake is Kondopozhskaya Bay, which makes up only 2.5% of the lake area. The bay experiences a long-term load of biogenic and organic matter from wastewater from a pulp and paper mill and trout farms [50,51,52,53]. According to the concentration of total phosphorus and chlorophyll *a*, Kondopozhskaya Bay is characterized as mesotrophic. In coastal areas of the bay experiencing anthropogenic pressure, these characteristics reach the level of eutrophic ecosystems (Table 4).

River runoff contains a large amount of organic matter, mainly of humus origin. The influence of rivers is felt in estuarine areas, where elevated values of TOC (6.6–8.8 mg L^−1^), watercolor (35–64 mg Pt-Co L^−1^), and TP—10–19 µg L^−1^ are observed [41,50].

### 5.2. Sampling

Water samples were taken from the surface water layer from aboard the Ecologist research vessel in July 2022. The set of samples (2 samples from each station) were collected from bloom spots in the southern Onego (station S_1, depth 32.5 m and station S_3, depth 34.0 m) and the Maloe Onego (st. L_5, depth 43.0 m), where there are no known sources of water pollution (Figure 4). A water sample was also taken from Kondopogskaya Bay, subjected to biogenic and organic load from the pulp and paper mills’ wastewater and the trout farms (station K50, depth 35 m).

### 5.3. Phytoplankton Analysis

To study phytoplankton, a 500 mL water sample was fixed with formalin at a final concentration of 1%, and then concentrated on a membrane filter with a pore diameter of 0.8 µm (Vladipor, Russia) to a sample volume of 5 mL above the filter, according to [54]. The cell density of cyanobacteria was determined using a standard counting chamber method following Guillard (1978) [55] and LeGresley and McDermott (2010) [56]. Biomass was assessed from individual cell volumes calculated from geometric figures [57]. Species making up over 10% of total abundance and biomass were assessed as dominant [58]. The term ‘bloom spot’ was used for conditions when the abundance of cyanobacteria in water samples is over 10^7^ cells L^−1^ [59]. The cyanobacterial species identification was performed according to [60,61]. The concentration of chlorophyll *a* in water samples was determined by spectrophotometric method [62].

### 5.4. HPLC–MSHR Analysis of Cyanotoxins

To distinguish water-dissolved and intracellular microcystin fractions and to separate planktonic DNA, water samples were divided into filtrate and phytoplankton biomass by filtration through Vladipor membrane filters (D pore = 0.8 µm). 

The profile and quantification of MCs were assessed by the method of high-performance liquid chromatography–mass spectrometry of high-resolution (HPLC–MSHR) using LTQ Orbitrap XL (Thermo Fisher Scientific, Waltham, MA, USA) with Prominence LC-20 system (Shimadzu, Japan). Sample preparation procedures included solid-phase extraction (Oasis HLB, Waters) for filtration and treatment with 75% methanol in an ultrasonic bath for biomass on filters.

MCs were separated chromatographically using Thermo Hypersil Gold RP C18 columns (100 × 3 mm, 3 µm, Thermo Fisher Scientific) in a gradient elution regime (0.2 mL min^−1^) by a water–acetonitrile mixture containing 0.05% formic acid. Mass-spectrometric analysis of MCs was performed under electrospray ionization conditions in a positive ion detection mode. Target compounds were identified by high-precision measurement of ion mass [M + H]^+^ or [M + 2H]^2+^ (resolution 30,000, accuracy within 5 ppm) [19], the collected fragmentation spectrum of the ion [22] and the retention time. Quantification was performed by the external standard method. Calibration dependencies were constructed using solutions of 9 standard MCs (MC–LR, MC–RR, MC–YR (Sigma Aldrich, St. Louis, MI, USA) and MC–LY, MC–LA, MC– LW, MC–LF, [D–Asp^3^]MC–LR и [D–Asp^3^]MC-RR (Enzo Life Sciences, Inc., Farmingdale, NY, USA). The procedural (taking into account concentrating stage during the sample preparation) LOD was 0.06 ng L^−1^, and LOQ was 0.2 ng L^−1^. Qualitative analysis for the presence of anatoxin-a was performed using high-resolution mass spectrometry. Isocratic elution (5% of organic phase) in the same chromatographic system was used.

### 5.5. DNA Extraction and PCR Detection of mcyA Gene

Polymerase chain reaction (PCR-analysis) was used to identify MC-producing cyanobacteria [63]. DNA from planktonic samples (environmental DNA, eDNA) was isolated with Diatom DNA Prep 200 reagents (OOO Isogene Laboratory, Russia) in accordance with the producer’s instructions. PCR amplification of MC biosynthesis *mcyA* gene was performed using the genus-specific primers mcyA_MF/MR (*Microcystis*, PCR product length is 225 bp) and mcyA AF/AR (*Dolichospermum*, 225 bp) [64]. Fragments of *mcyA* gene were amplified in a CFX96 Touch thermocycler (Bio–Rad, Hercules, CA, USA) using 20 µL of GenPak PCR Core ready-made mixture of reagents (Isogene Laboratory, Moscow, Russia) in accordance with the following protocol: preliminary DNA denaturation at 95 °C for 3 min followed by 40 amplification cycles at 95 °C for 30 s, 58 °C for 30 s and 72 °C for 1 min and the last stage (elongation) at 72 °C for 10 min. DNAs from the MC-producing strain *Microcystis aeruginosa* PCC 7806 and field colonies of *Dolichospermum lemmermannii* [65] were used as positive controls. PCR products were fractionated electrophoretically in 1.5% agar gel and analyzed in UV light after staining with ethidium bromide using the Gel Doc XR+ gel-documenting system (Bio–Rad, Hercules, CA, USA).

## Figures and Tables

**Figure 1 toxins-16-00457-f001:**
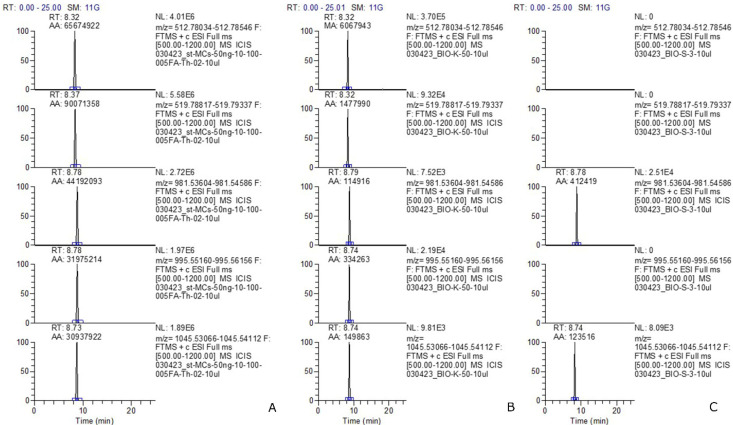
Extracted ion chromatogram of high resolution (mass accuracy within 5 ppm) for MC congeners detected in phytoplankton samples of Lake Onego: standard mix solution, concentration 50 ng mL^−1^ (**A**), sample K_50 (**B**), sample S_3 (**C**) (from top to bottom): *m*/*z* 512.78235 ([M+2H]^2+^ [D-Asp^3^]MC-RR); *m*/*z* 519.79077 ([M+2H]^2+^ MC-RR); *m*/*z* 981.54095 ([M+H]^+^ [D-Asp^3^]MC-LR); *m*/*z* 995.55658 ([M+H]^+^ MC-LR); *m*/*z* 1045.53589 ([M+H]^+^ MC-YR).

**Figure 2 toxins-16-00457-f002:**
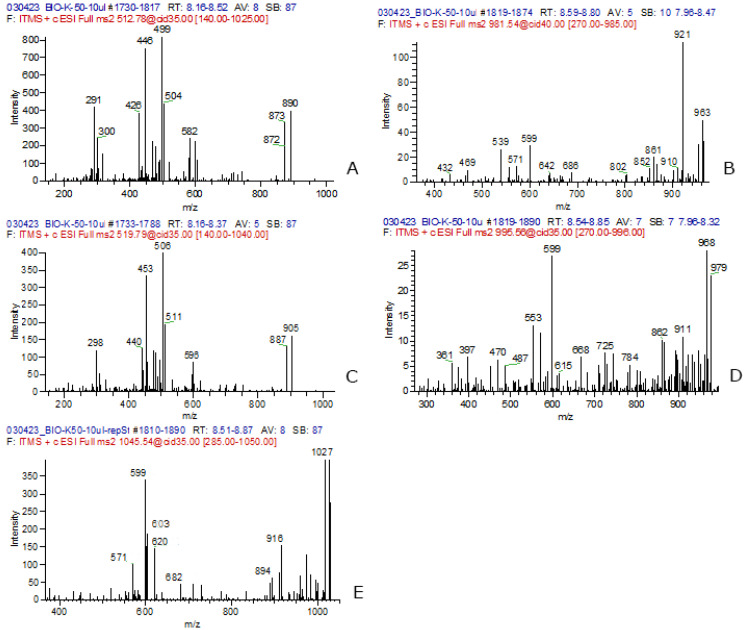
MS2 spectra of the MC congeners identified in the extract of sample K_50: [D-Asp^3^]MC-RR) (**A**); [D-Asp^3^]MC-LR) (**B**); MC-RR (**C**); MC-LR (**D**); MC-YR (**E**).

**Figure 3 toxins-16-00457-f003:**
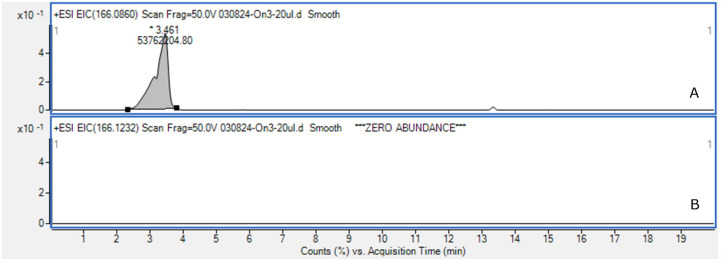
Example of extracted ion high-resolution chromatogram (mass accuracy within 5 ppm) of phytoplankton extract (sample S_3): phenylalanine (**A**), anatoxin-a (**B**).

**Figure 4 toxins-16-00457-f004:**
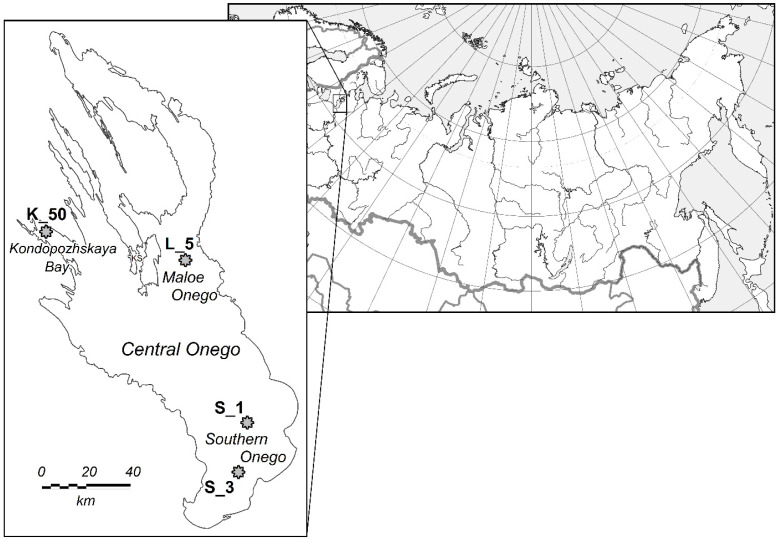
Location of sampling stations. Lake Onego, July 2022.

**Table 1 toxins-16-00457-t001:** Abundance (N_tot_) and biomass (B_tot_) of the cyanobacterial community and chlorophyll *a* in water samples from Lake Onego in July 2022.

Station	N_tot_, 10^6^ cells L^−1^	B_tot_, mg L^−1^	Chl *a*, µg L^−1^
S_1	129.29	36.42	31
S_3	84.52	19.38	33
L_5	24.32	6.82	47
K50	12.23	0.69	11

**Table 2 toxins-16-00457-t002:** Species composition and structure of the cyanobacterial community in water samples from Lake Onego in July 2022.

Sampling Station	Species	Share of the Species in Abundance, %	Share of the Species in Biomass, %
S_1	*Aphanizomenon flos-aquae*	0.003	0.04
	*Dolichospermum lemmermannii*	57	74.9
	*Dolichospermum spiroides*	43	25.0
S_3	*Aphanizomenon flos-aquae*	0.05	0.79
	*Aphanothece minutissima*	1	0.04
	*Aphanocapsa grevillei*	1	0.06
	*Dolichospermum circinale*	49	31.2
	*Dolichospermum lemmermannii*	46	67.7
	*Microcystis aeruginosa*	2	0.12
L_5	*Aphanothece nidularis*	8	0.3
	*Dolichospermum circinale*	44	28
	*Dolichospermum lemmermannii*	49	71.7
K50	*Aphanizomenon flos-aquae*	0.007	16.5
	*Dolichospermum flos-aquae*	56	52.6
	*Dolichospermum spiroides*	12	11.5
	*Dolichospermum lemmermannii*	4	8.7
	*Microcystis aeruginosa*	26	10.6

**Table 3 toxins-16-00457-t003:** Intracellular MC concentration (ng L^−1^) and the results of PCR detection of the *mcyA* gene in samples from Lake Onego in July 2022.

MC Congeners and Detection of *mcyA* Gene	S_1	K50	S_3	L_50
[D-Asp^3^]MC-LR	‒ ^1^	2	8	‒
MC-LR	‒	8	‒	‒
[D-Asp^3^]MC-RR	‒	125	‒	‒
MC-RR	‒	14	‒	‒
MC-YR	‒	4	4	‒
Total MCs concentration	‒	153	12	‒
*Dolichospermum*-specific *mcyA*	‒	+ ^2^	+	NA
*Microcystis*-specific *mcyA*	+	+	+	NA

^1^ “‒” MCs (< 0.06 ng L^−1^) or the *mcyA* gene were not detected, ^2^ “+” the *mcyA* gene was detected in eDNA, NA—not analyzed.

**Table 4 toxins-16-00457-t004:** Averaged chemical variables of Onelo Lake water in 2019–2021 [34,43].

Characteristics	Pelagic zone of Lake Onego	Kondopozhskaya Bay Off-Shore	Kondopozhskaya Bay In-Shore
Conductivity, μS cm^−1^	50.7	48.5	53.0
pH	7.3	7.2	7.0
TOC, mg L^−1^	6.4	7.6	12.3
Color, mg Pt-Co L^−1^	31.0	35.6	51.7
TP, µg L^−1^	8.1	22.0	68.5
PO4-P, µg L^−1^	0	0	19.0
NO3-N, mg L^−1^	0.15	0.16	0.10
NH4-N, mg L^−1^	0.01	0.01	0.08
TN, mg L^−1^	0.36	0.38	0.56
Chlorophyll *a*, µg L^−1^	3.2	4.6	8.8

## Data Availability

Data supporting the presented results are available in the text of the article and in the Appendix A.

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
