# Peer review of "Toxigenic Cyanobacteria and Microcystins in a Large Northern Oligotrophic Lake Onego, Russia"

_toxins, 2024, doi:10.3390/toxins16110457_

Round 1
Reviewer 1 Report
Comments and Suggestions for Authors
General comments
This manuscript present results from a survey on cyanobacteria and MC production monitoring in Lake Onego, a large northern oligotrophic lake, most of it in a near pristine state. The geographical location of the survey is interesting, since it corresponds to underexplored areas and singular habitats.
Contrary to what could be expected, the higher biomass of potentially toxigenic cyanobacteria was detected in the sampling stations with no sources of pollution (characterized as oligotrophic). Despite its lower total cyanobacteria abundance, the polluted station (characterized as mesotrophic) showed higher values of MC per cell. This could be probably due to the presence at higher levels in that station of Microcystis aeruginosa, which, as the authors discuss seems to have more effective production of MC. The presence of this species in the mesotrophic station and its almost total absence in the oligotrophic ones makes sense. The authors discuss their findings in the proper context. However, I miss some a priori hypothesis about the distribution of cyanobacteria and MC across the stations, and also some possible explanation for the lower Chl a content of the mesotrophic polluted station.
The results are interesting in the sense that show how toxigenic blooms could reach levels dangerous to human health even in these relatively cold and high latitude waters. At the same time, it is interesting to point out that, globally, toxicity was higher in the non-polluted areas, whereas the polluted areas, the higher presence of M. aeruginosa could turn them more dangerous in the future (if the sources of pollution persist) or in other times of the year.
Being all the above said, since the sampling was done in a single date (22 July) the conclusions would not be that relevant if this date does not correspond, at least approximately, with the expected peak of cyanobacterial biomass. Authors need to clarify this in order to give a proper context for the robustness of the conclusions.
Specific comments
- Title: “a largest” is wrong, should be “the largest” (I don’t know if it is the case) or simply “a large”.
- Line 27: I suggest “Cyanobacterial bloms are …”
- Lines 42-43: I suggest “is missing” not “is lack”.
- Line 170: I suggest “one order of magnitude”.
- Line 174: I suggest “a similar observation”.
- Material and Methods. 5.2 Sampling: the number of samples should be detailed here.
Author Response
We are grateful for the positive evaluation of our manuscript.
Our answers are below:
General comments
The Reviewer: This manuscript present results from a survey on cyanobacteria and MC production monitoring in Lake Onego, a large northern oligotrophic lake, most of it in a near pristine state. The geographical location of the survey is interesting, since it corresponds to underexplored areas and singular habitats. Contrary to what could be expected, the higher biomass of potentially toxigenic cyanobacteria was detected in the sampling stations with no sources of pollution (characterized as oligotrophic). Despite its lower total cyanobacteria abundance, the polluted station (characterized as mesotrophic) showed higher values of MC per cell. This could be probably due to the presence at higher levels in that station of Microcystis aeruginosa, which, as the authors discuss seems to have more effective production of MC. The presence of this species in the mesotrophic station and its almost total absence in the oligotrophic ones makes sense. The authors discuss their findings in the proper context. However, I miss some a priori hypothesis about the distribution of cyanobacteria and MC across the stations, and also some possible explanation for the lower Chl a content of the mesotrophic polluted station. The results are interesting in the sense that show how toxigenic blooms could reach levels dangerous to human health even in these relatively cold and high latitude waters. At the same time, it is interesting to point out that, globally, toxicity was higher in the non-polluted areas, whereas the polluted areas, the higher presence of M. aeruginosa could turn them more dangerous in the future (if the sources of pollution persist) or in other times of the year.
Answer: We added the sentence (Lines 73-75) with explanation of low chlorophyll a in the mesotrophic bay. “. In Kondopozhskaya Bay, the concentration of chlorophyll a was also 3–4 times lower, which was associated with the recorded low abundance and biomass (Table 1)”
During the study, we found an atypical pattern of spatial distribution of cyanobacteria and Chl a in the reservoir. It is known that local areas with high cyanobacterial biomass are formed in the coastal part of the lake and in its bays, where the level of biogenic loading is higher. However, it has been established that the studied lake is characterized by large-scale horizontal circulation water currents (Rukhovets and Filatov, 2010). Thus, we assumed that the low concentration of Chl a and cyanobacterial biomass in the mesotrophic bay compared to the oligotrophic pelagic zone could be explained by the transfer of bloom spots by currents from more polluted parts of the lake (coastal zone and bays) to the oligotrophic pelagic zone. However, to test this hypothesis, more detailed data on the spatial distribution of cyanobacteria in the lake area are needed than those obtained in our first screening study of toxigenic cyanobacteria in Lake Onego.
We have briefly described our explanation and added it to the discussion (Lines 180-195):
“We hypothesize that bloom spots formed in the coastal zone and in the bays can subsequently be carried by the current to the pelagic zone. Moreover, it is known that the Lake Onego has a system of stable vigorous circulation currents [Rukhovets and Filatov, 2010]. A similar mechanism of surface scum formation in the pelagic zone has been noted in other large oligotrophic lakes. For instance, bloom spot of Dolichospermum in Lake Superior moved over a distance of up to 8 km from the shore [Sterner et al., 2020]”.
The Reviewer: Being all the above said, since the sampling was done in a single date (22 July) the conclusions would not be that relevant if this date does not correspond, at least approximately, with the expected peak of cyanobacterial biomass. Authors need to clarify this in order to give a proper context for the robustness of the conclusions.
Answer: The discussion provided information that, according to literature data and long-term personal observations, the seasonal maximum of Dolichospermum development in Lake Onego is usually recorded from late July to August.
Lines 146-147: “The seasonal maximum of Dolichospermum in Lake Onego usually occurs in late July‒August in a period of maximal water warming. Until now, the abundance of D. flos-aquae and D. lemmermannii in the pelagic zone did not exceed 103 cells L-1 [20,21]”.
Phytoplankton samples were collected on July 22 based on the collected data on the seasonal maximum of cyanobacterial biomass during this period.
Specific comments
- Title: “a largest” is wrong, should be “the largest” (I don’t know if it is the case) or simply “a large”. Corrected.
- Line 27: I suggest “Cyanobacterial bloms are …” Corrected
- Lines 42-43: I suggest “is missing” not “is lack”. Corrected
- Line 170: I suggest “one order of magnitude”. Corrected. Now it is Line 186
- Line 174: I suggest “a similar observation”. Corrected. Now it is Line 196
- Material and Methods. 5.2 Sampling: the number of samples should be detailed here. We added the information in Line 281.
Reviewer 2 Report
Comments and Suggestions for Authors
In this paper, the authors report the detection of toxigenic cyanobacteria and microcystins in the oligotrophic pelagic zone and mesotrophic bay of Lake Onego. They identify cyanobacterial species such as Dolichospermum and Microcystis, finding that blooms in both zones pose potential hazards to humans and animals. Despite detecting microcystins (e.g., MC-LR) in trace amounts, the study discusses the low cyanotoxin content in phytoplankton dominated by Dolichospermum.
The paper provides an important study on cyanobacterial blooms and cyanotoxin presence in the oligotrophic Lake Onego. However, there are several aspects of the study that should be reconsidered:
1. The introduction and discussion do not adequately address the broader ecological and economic impacts of HABs on fisheries and public health. While they mention the occurrence of cyanotoxins and their hazards, there's no discussion on bioaccumulation in fish, a significant pathway for toxins entering human food chains. This is particularly surprising given the increasing literature on bioaccumulation, including recent studies showing hepatotoxins accumulating in fish tissues in lakes with similar blooms. Ignoring these impacts limits the relevance of the study for policymakers and public health officials. Some references to leverage could include the following:
World Health Organization guidelines:
-Chorus, I., & Welker, M. (Eds.). (2021). Toxic cyanobacteria in water: a guide to their public health consequences, monitoring and management. CRC Press, Boca Raton (FL), on behalf of the World Health Organization, Geneva, CH. https://doi.org/10.1201/9781003081449
Socioeconomic costs of HABs:
-Smith, R. B., Bass, B., Sawyer, D., Depew, D., & Watson, S. B. (2019). Estimating the economic costs of algal blooms in the Canadian Lake Erie Basin. Harmful Algae, 87, 101624. https://doi.org/10.1016/j.hal.2019.101624
Geographical spread of Microcystis:
-Harke, M. J., Steffen, M. M., Gobler, C. J., Otten, T. G., Wilhelm, S. W., Wood, S. A., & Paerl, H. W. (2016). A review of the global ecology, genomics, and biogeography of the toxic cyanobacterium, Microcystis spp. Harmful algae, 54, 4-20. https://doi.org/10.1016/j.hal.2015.12.007
Impacts on wildlife populations:
-de Magalhães, V. F., Soares, R. M., & Azevedo, S. M. (2001). Microcystin contamination in fish from the Jacarepaguá Lagoon (Rio de Janeiro, Brazil): ecological implication and human health risk. Toxicon, 39(7), 1077-1085. https://doi.org/10.1016/S0041-0101(00)00251-8
-Poste, A. E., Hecky, R. E., & Guildford, S. J. (2011). Evaluating microcystin exposure risk through fish consumption. Environmental science & technology, 45(13), 5806-5811. https://doi.org/10.1021/es200285c
-Onyango, D. M., Orina, P. S., Ramkat, R. C., Kowenje, C., Githukia, C. M., Lusweti, D., & Lung'ayia, H. B. (2020). Review of current state of knowledge of microcystin and its impacts on fish in Lake Victoria. Lakes & Reservoirs: Research & Management, 25(3), 350-361. https://doi.org/10.1111/lre.12328
-Shahmohamadloo, R. S., Bhavsar, S. P., Almirall, X. O., Marklevitz, S. A., Rudman, S. M., & Sibley, P. K. (2023a). Lake Erie fish safe to eat yet afflicted by algal hepatotoxins. Science of the Total Environment, 861, 160474. https://doi.org/10.1016/j.scitotenv.2022.160474
-Shahmohamadloo, R. S., Bhavsar, S. P., Almirall, X. O., Marklevitz, S. A., Rudman, S. M., & Sibley, P. K. (2023b). Cyanotoxins accumulate in Lake St. Clair fish yet their fillets are safe to eat. Science of the Total Environment, 874, 162381. https://doi.org/10.1016/j.scitotenv.2023.162381
-Shahmohamadloo, R. S., Frenken, T., Rudman, S. M., Ibelings, B. W., & Trainer, V. L. (2023c). Diseases and disorders in fish due to harmful algal blooms. In Climate Change on Diseases and Disorders of Finfish in Cage Culture (pp. 387-429). GB: CABI. https://doi.org/10.1079/9781800621640.0010
2. The study mentions degradation of aquatic ecosystems and trophic simplification but does not investigate deeply into how cyanobacterial blooms disrupt food webs, affecting both the structure and function of aquatic ecosystems. There is ample research showing that blooms not only reduce biodiversity but also affect the health of fish populations by altering oxygen levels, leading to fish kills, and impacting species composition. The above references can help the authors in their revisions.
3. The study provides a snapshot of cyanobacterial and microcystin concentrations at a few locations in Lake Onego during a single month. While the findings are valuable, the paper lacks long-term or seasonal data that would help in understanding how blooms develop, peak, and decline over time. For instance, it would have been helpful to know whether the detected blooms are a recurring problem or a single, anomalous event. A broader sampling regime would have provided more robust insights into bloom dynamics. Also, there’s a lack of description about the aquatic life of Lake Onego. Tell the reader more what benefits this lake brings (e.g., ecosystem services). It puts it into broader context why it matters to investigate this issue.
Specific line comments:
-L53-64: Can you acquire the meteorological data and tell us what the temperatures were like? This will help us understand why it was “abnormally high”.
-Reference 24 is outdated (Chorus). Please update it with the 2021 reference I have shared above.
-L165-190: It is difficult to draw these conclusions if regular water samples were not collected both temporally and spatially, and across the water column.
Author Response
Dear Reviewer 1 !
We thank you for your positive assessment of our work and useful suggestions!
Our answers to your comments are below:
Comments and Suggestions for Authors
Reviewer: 1. The introduction and discussion do not adequately address the broader ecological and economic impacts of HABs on fisheries and public health. While they mention the occurrence of cyanotoxins and their hazards, there's no discussion on bioaccumulation in fish, a significant pathway for toxins entering human food chains. This is particularly surprising given the increasing literature on bioaccumulation, including recent studies showing hepatotoxins accumulating in fish tissues in lakes with similar blooms. Ignoring these impacts limits the relevance of the study for policymakers and public health officials.
The answer: Based on your recommendations, we have added general information about the broader impact of blooms on large oligotrophic lake ecosystems, including fish, to the introduction. Thank you for the suggested links. However, in this work, we did not plan to study and deeply analyze the negative impact of cyanobacterial blooms on fish and disruption of food chains. This is a very interesting area of research and certainly useful. However, the purpose of our article was to obtain the first data on the occurrence and distribution of potentially toxic cyanobacteria and microcystins in a large cold oligotrophic Lake Onego.
Reviewer: Some references to leverage could include the following:
World Health Organization guidelines: Chorus, I., & Welker, M. (Eds.). (2021). Toxic cyanobacteria in water: a guide to their public health consequences, monitoring and management. CRC Press, Boca Raton (FL), on behalf of the World Health Organization, Geneva, CH. https://doi.org/10.1201/9781003081449
It was mentioned as Ref.31 in the first MS version, now it is Ref.36
Socioeconomic costs of HABs: -Smith, R. B., Bass, B., Sawyer, D., Depew, D., & Watson, S. B. (2019). Estimating the economic costs of algal blooms in the Canadian Lake Erie Basin. Harmful Algae, 87, 101624. https://doi.org/10.1016/j.hal.2019.101624
We added it, now it is Ref.15
Geographical spread of Microcystis: Harke, M. J., Steffen, M. M., Gobler, C. J., Otten, T. G., Wilhelm, S. W., Wood, S. A., & Paerl, H. W. (2016). A review of the global ecology, genomics, and biogeography of the toxic cyanobacterium, Microcystis spp. Harmful algae, 54, 4-20. https://doi.org/10.1016/j.hal.2015.12.007
It was mentioned as Ref 6 in the first MS version
Impacts on wildlife populations: de Magalhães, V. F., Soares, R. M., & Azevedo, S. M. (2001). Microcystin contamination in fish from the Jacarepaguá Lagoon (Rio de Janeiro, Brazil): ecological implication and human health risk. Toxicon, 39(7), 1077-1085. https://doi.org/10.1016/S0041-0101(00)00251-8
Poste, A. E., Hecky, R. E., & Guildford, S. J. (2011). Evaluating microcystin exposure risk through fish consumption. Environmental science & technology, 45(13), 5806-5811. https://doi.org/10.1021/es200285c
-Onyango, D. M., Orina, P. S., Ramkat, R. C., Kowenje, C., Githukia, C. M., Lusweti, D., & Lung'ayia, H. B. (2020). Review of current state of knowledge of microcystin and its impacts on fish in Lake Victoria. Lakes & Reservoirs: Research & Management, 25(3), 350-361. https://doi.org/10.1111/lre.12328
-Shahmohamadloo, R. S., Frenken, T., Rudman, S. M., Ibelings, B. W., & Trainer, V. L. (2023c). Diseases and disorders in fish due to harmful algal blooms. In Climate Change on Diseases and Disorders of Finfish in Cage Culture (pp. 387-429). GB: CABI. https://doi.org/10.1079/9781800621640.0010
These above listed references were added (Ref.11-14)
-Shahmohamadloo, R. S., Bhavsar, S. P., Almirall, X. O., Marklevitz, S. A., Rudman, S. M., & Sibley, P. K. (2023a). Lake Erie fish safe to eat yet afflicted by algal hepatotoxins. Science of the Total Environment, 861, 160474. https://doi.org/10.1016/j.scitotenv.2022.160474
-Shahmohamadloo, R. S., Bhavsar, S. P., Almirall, X. O., Marklevitz, S. A., Rudman, S. M., & Sibley, P. K. (2023b). Cyanotoxins accumulate in Lake St. Clair fish yet their fillets are safe to eat. Science of the Total Environment, 874, 162381. https://doi.org/10.1016/j.scitotenv.2023.162381
These two references were added (Ref.37,38)
Reviewer. 2. The study mentions degradation of aquatic ecosystems and trophic simplification but does not investigate deeply into how cyanobacterial blooms disrupt food webs, affecting both the structure and function of aquatic ecosystems. There is ample research showing that blooms not only reduce biodiversity but also affect the health of fish populations by altering oxygen levels, leading to fish kills, and impacting species composition. The above references can help the authors in their revisions.
Answer: You propose to conduct a very interesting study, but this is not the purpose of our work. For this kind of research, it is necessary to accumulate a larger volume of initial data. In the future, we are planning to conduct it
Reviewer: 3. The study provides a snapshot of cyanobacterial and microcystin concentrations at a few locations in Lake Onego during a single month. While the findings are valuable, the paper lacks long-term or seasonal data that would help in understanding how blooms develop, peak, and decline over time. For instance, it would have been helpful to know whether the detected blooms are a recurring problem or a single, anomalous event. A broader sampling regime would have provided more robust insights into bloom dynamics. Also, there’s a lack of description about the aquatic life of Lake Onego. Tell the reader more what benefits this lake brings (e.g., ecosystem services). It puts it into broader context why it matters to investigate this issue.
Answer: The paragraph about benefits this lake brings was added (lines 134-138).
In the future, we are planning to conduct a long time research and study the cyanotoxins distribution in time and throughout the lake.
Specific line comments:
-L53-64: Can you acquire the meteorological data and tell us what the temperatures were like? This will help us understand why it was “abnormally high”.
Answer: Now these are lines 59-60 “The total abundance and biomass of cyanobacteria in the pelagic zone were abnormally high for these oligotrophic parts (Table 1)”. The words “abnormally high” were corresponded to the abundance and biomass values, and these data were presented in the Table 1.
Reviewer: Reference 24 is outdated (Chorus). Please update it with the 2021 reference I have shared above.
Answer: We used both references in the discussion. Reference Сhorus, 2021 was also present in our manuscript (reference 31 in the first version, reference 36 in the corrected version). The 1999 work used cyanobacteria abundance values as an indicator of potential hazard to humans due to the probable presence of microcystins. We made this assessment in lines 148-152 "In our study, the recorded cyanobacteria abundance in the samples from pelagic zone of Lake Onego exceeded 20–100 × 106 cells L-1 (Table 2). In accordance with the WHO guidelines, the recorded abundance of cyanobacteria is associated with the expected high concentration of MC, which may pose a risk to human health [29]". The 2021 work already proposes to use other parameters (chlorophyll a and cyanobacterial biomass) for risk assessment. Therefore, we cannot remove the reference Chorus,1999
Reviewer: L165-190: It is difficult to draw these conclusions if regular water samples were not collected both temporally and spatially, and across the water column.
Answer: In this paragraph we only put forward 2 hypotheses to explain the low intracellular MCs content in surface scum dominated by Dolichospermum. In support of this, we cite literature data on lakes from Germany and Denmark, as well as information on the discovery of genes for the biosynthesis of other cyanotoxins in Dolichospermum akinetes, but not microcystins! Moreover, the second hypothesis about akinetes is put forward in the scientific literature on this issue for the first time. In order to further test these hypotheses, we plan to take several samples from a similar bloom of water in the lake and isolate Dolichospermum strains and study the production of toxins by a separate strain from Lake Onego.
Reviewer 3 Report
Comments and Suggestions for Authors
Dear authors and editors,
The manuscript Toxigenic cyanobacteria and microcystins in a largest northern oligotrophic lake Onego, Russia presents new and valuable data on the occurrence of microcystins and their potential producers in the oligotrophic pelagic zone and mesotrophic bay of Lake Onego –the second largest freshwater lake in Europe. As this lake is mentioned several times in the text, mostly in different ways, and with the aim of improving the final version of the manuscript, I have a suggestion for a small correction:
The authors wrote: in the title - largest northern oligotrophic lake Onego;
…in the abstract - Lake Onego –the second largest lake in Europe;
…in the key contribution - the oligotrophic Lake Onego, the second largest lake in Europe;
…in the introduction - the northern large oligotrophic Lake Onego;
….in the results - oligotrophic cold-water Lake Onego;
…in the conclusions - the large northern oligotrophic Lake Onego;
…in the materials and methods - Lake Onego, the second largest body of water in Europe.
In my opinion, the full description of the lake - location, trophic status, size, etc. - should be included in the Materials and Methods chapter. All other mentions in the text can only be replaced by Lake Onego, as all other different descriptions will mislead the reader.
Author Response
Dear Reviewer 2!
Thank you for your positive assessment of our work.
Reviewer’s Comment:
“As this lake is mentioned several times in the text, mostly in different ways, and with the aim of improving the final version of the manuscript, I have a suggestion for a small correction:
The authors wrote: in the title - largest northern oligotrophic lake Onego;
…in the abstract - Lake Onego –the second largest lake in Europe;
…in the key contribution - the oligotrophic Lake Onego, the second largest lake in Europe;
…in the introduction - the northern large oligotrophic Lake Onego;
….in the results - oligotrophic cold-water Lake Onego;
…in the conclusions - the large northern oligotrophic Lake Onego;
…in the materials and methods - Lake Onego, the second largest body of water in Europe.
In my opinion, the full description of the lake - location, trophic status, size, etc. - should be included in the Materials and Methods chapter. All other mentions in the text can only be replaced by Lake Onego, as all other different descriptions will mislead the reader.”
The answer:
Regarding your comments, we would like to clarify that by using various characteristics of the lake when naming it, we wanted to emphasize that the same observations can also be made in lakes with similar characteristics (cold-water, oligotrophic and large). However, we have removed these extended names of the lake further in the text.
Round 2
Reviewer 2 Report
Comments and Suggestions for Authors
The authors have thoughtfully and carefully revised the manuscript. It is sufficient and acceptable for publication.